# Polyacrylonitrile/Aminated Polymeric Nanosphere Nanofibers as Efficient Adsorbents for Cr(VI) Removal

**DOI:** 10.3390/molecules27207133

**Published:** 2022-10-21

**Authors:** Junwen Qi, Mengli Zeng, Zhigao Zhu, Yujun Zhou, Xiuyun Sun, Jiansheng Li

**Affiliations:** Jiangsu Key Laboratory of Chemical Pollution Control and Resources Reuse, School of Environmental and Biological Engineering, Nanjing University of Science and Technology, Nanjing 210094, China

**Keywords:** aminated polymeric nanospheres, nanofibers, Cr(VI), adsorption, redox reaction

## Abstract

In this work, polyacrylonitrile/aminated polymeric nanosphere (PAN/APN) nanofibers were prepared by electrospinning of monodispersed aminated polymeric nanospheres (APNs) for removal of Cr(VI) from aqueous solution. Characterization results showed that obtained PAN/APNs possessed nitrogen functionalization. Furthermore, the adsorption application results indicated that PAN/APN nanofibers exhibited a high adsorption capacity of 556 mg/g at 298 K for Cr(VI) removal. The kinetic data showed that the adsorption process fits the pseudo-second order. A thermodynamic study revealed that the adsorption of Cr(VI) was spontaneous and endothermic. The coexisting ions Na^+^, Ca^2+^, K^+^, Cl^−^, NO_3_^−^ and PO_4_^3−^ had little influence on Cr(VI) adsorption, while SO_4_^2−^ in solution dramatically decreased the removal performance. In the investigation of the removal mechanism, relative results indicated that the adsorption behavior possibly involved electrostatic adsorption, redox reaction and chelation. PAN/APN nanofibers can detoxify Cr(VI) to Cr(III) and subsequently chelate Cr(III) on its surface. The unique structure and nitrogen functionalization of PAN/APN nanofibers make them novel and prospective candidates in heavy metal removal.

## 1. Introduction

In the past few years, the persistence of extremely toxic heavy metals in water has attracted much attention [1]. Due to their toxicity and low biodegradation, heavy metals not only affect the health of human beings and aquatic life but also cause serious environmental pollution [2,3]. Among the various heavy metals, chromium (Cr), which is extensively used in steel fabrication, nuclear power plants, electroplating, refractories, pigments, tanning industries and chemical manufacturing [4], is on the top-priority list of toxic pollutants defined by agencies of environmental protection in different countries [5,6]. According to the National Standards in China, the maximum discharges of total chromium from municipal wastewater treatment plants and the electronic industry are 0.1 and 1.0 mg/L, respectively. Therefore, the removal of chromium in industrial wastewater is increasingly urgent.

Under natural circumstances, there are two common oxidation statuses of chromium, i.e., Cr(III) and Cr(VI). Owing to the high mobility and solubility of Cr(VI), larger uptake was found for cell membranes in aquatic conditions. In addition, Cr(VI) is hundreds of times more toxic than Cr(III) [7]. Various technologies have been applied to remove Cr(VI), such as ion exchange, reverse osmosis, electrolytic, precipitation, flocculation and adsorption [8,9]. Compared with other methods, adsorption has been widely used to remove Cr(VI) from wastewater due to its cost-effectiveness, ease in handling and high performance [10].

Several classes of adsorbents have been developed for Cr(VI) removal, including activated carbon, graphene, carbon nanotubes, polymeric materials, metal oxide nanoparticles, zeolite and silica [11,12,13]. Among them, organic nanomaterials are prospective candidates due to their abundant functional groups and nanosized structures which could highly coordinate and enhance adsorption capacity [14,15]. Considering the high toxicity of Cr(VI), the synergetic reduction–adsorption strategy is more desirable [16,17,18]. In particular, amine functional groups can synchronously reduce Cr(VI) to Cr(III) and then adsorb Cr(III) from contaminated water. Amine functionalization was confirmed on MCM-41 and graphene oxide via grafting and impregnation [19,20]. The obtained amine-functionalized absorbent exhibited high adsorption capacity and good reduction ability. However, the limited reusability was a normal issue on these adsorbents prepared via post-modification. Moreover, due to the powder nature, they usually suffered from complicated separation procedures which would increase operation costs [21]. Hence, it is still urgent to develop macroscale amine functional polymeric absorbents to avoid complicated separation process.

With the development of electrospinning technology, flexible one-dimensional fibers provide an opportunity to solve the aggregation and separation problem during the application of nanoadsorbents [22,23]. Hota et al. showed effective removal of heavy metals by Fe_2_O_3_-Al_2_O_3_ fiber obtained through a sol–gel electrospinning method [24]. Aliabadi et al. fabricated nanofibrous PET/PAN/GO/Fe_3_O_4_ for heavy metal removal [25]. Pan et al. reported the development of a polyacrylonitrile nanofiber mat, PAN/PPy/MnO_2_, for the removal of lead [26]. Nanofibers prepared by electrospinning exhibited high surface area, abundant functional groups and easy operability, thus representing promising adsorbents.

Motivated by the above works, the electrospinning technique was introduced in the formation of monodispersed aminated polymeric nanospheres (APNs) to prepare polyacrylonitrile/aminated polymeric nanosphere (PAN/APN) nanofibers. Their morphological structure and chemical properties were characterized. Cr(VI) was used as the representative of heavy metals to evaluate the adsorption and reduction performance in detail. Furthermore, the breakthrough behavior in a fixed-bed column experiment was assessed. The removal mechanism was carefully investigated through various methods.

## 2. Materials and Methods

### 2.1. Reagents

Ethanol, HCl (37 wt%), KOH, K_2_Cr_2_O_7_, ethylenediamine (EDA) and HCHO (37 wt%) were obtained from Nanjing Chemical Reagent Co., Ltd. N,N-dimethylformamide (DMF) and resorcinol were purchased from the Sinopharm Chemical Reagent Co., Ltd. Polyacrylonitrile (PAN) and polyvinyl pyrrolidone (PVP) were provided by Sigma-Aldrich. Deionized water (18.2 MΩ) was used in all experiments.

### 2.2. Synthesis of APNs

As previous reported [27], 0.274 mol (16 mL) ethanol, 2.216 mol (40 mL) deionized water and 8.7 mol (0.6 mL) EDA were mixed under stirring. Then, 3.6 mmol (0.4 g) resorcinol was added and stirred for another 30 min. Then, 0.6 mL HCHO (37 wt%) was added dropwise into the above mixture and stirred for 24 h at 30 °C. Finally, aminated polymeric nanospheres (APNs) were obtained by centrifuging and dried at 100 °C for 24 h.

### 2.3. Synthesis of PAN/APN and PAN Nanofibers

PAN/APN nanofibers were prepared via an electrospinning process [28] with modifications as follows: 1 g APNs, 0.1 g PVP and 0.5 g PAN were dispersed and dissolved in 64.6 mmol (5 mL) DMF. The solution was constantly stirred for 8 h under 70 °C and then loaded into a 10 mL syringe fitted with a metallic needle. A high voltage of 12 kV was applied to the metallic needle as the positive terminal, and the negative terminal was connected to the grounded collector. The obtained nanofibers were polyacrylonitrile/aminated polymeric nanosphere (PAN/APN) nanofibers. PAN nanofibers were prepared for comparison as mentioned above without APNs.

### 2.4. Characterization

Transmission electron microscopy (TEM) analysis was conducted on a TECNAI G2 20 electron microscope operated at 200 kV. Scanning electron microscopy (SEM) analysis was conducted on FEI Quanta 250 F system. Nitrogen adsorption–desorption isotherms were analyzed using a Micromeritics ASAP-2020 at 77 K. X-ray photoelectron spectroscopy (XPS) spectra were obtained using a PHI Quantera II ESCA system with Al Kα radiation at 1486.8 eV. Fourier transform infrared spectroscopy (FTIR) was performed using a Prestige 21 at 400–4000 cm^−1^. The zeta potential of APNs was analyzed using a Brookhaven zeta potential analyzer. Concentrations of chromium were analyzed via atomic absorption spectroscopy (AAS) and ultraviolet and visible spectroscopy (UV-Vis), which were conducted on a Perkin Elmer PinAAcle 900 and Lambda 750, respectively.

### 2.5. Adsorption Experiment

The Cr(VI) adsorption capacity of the adsorbent was evaluated in a 250 mL Erlenmeyer flask by batch equilibrium method. Nanofibers were added to the flask with a Cr(VI) aqueous solution, which was shaken violently (180 rpm) at 25 °C for a given time. To research the effect of pH on the adsorption processes, the solution pH throughout the experiment was adjusted using HCl and KOH. In the kinetic study, the concentration of Cr(VI) was 50 mg/L, and 1 mL solution was sampled at various time intervals. Adsorption isotherm experiments were conducted by adding the same quantity of adsorbents to a series of conical flasks that contained Cr(VI) solutions with different concentrations (50–500 mg/L). A certain amount of ions (Na^+^, Ca^2+^, K^+^, SO_4_^2−^, Cl^−^, NO_3_^−^, PO_4_^3−^) was added into the aqueous solution to study the competing adsorption behaviors. Fixed-bed column experiments were also conducted. The bed volume of PAN/APN nanofibers was set as 0.78 cm^3^. The initial concentration and pH of Cr(VI) wastewater were fixed at 10 mg/L and 2, respectively. The flow rate was 30 BV/h. In the whole experimental section, total chromium concentrations were measured by AAS, and concentrations of Cr(VI) were measured by UV-Vis at 540 nm. The adsorption experiments and concentration tests were performed in triplicate to guarantee the reproducibility of data.

### 2.6. Modeling

The equilibrium of adsorption capacity (*Q_e_*, mg/g) was estimated by (Equation (1)):(1)$$Q_{e}=\frac{(C_{0}-C_{e})\times V}{m}$$
where *C_0_* (mg/L) is the initial concentration of Cr(VI) and *C_e_* (mg/L) is the equilibrium concentration. *V* (mL) is the volume of the solution and *m* (g) is the dose of absorbent. Adsorption isotherms were obtained by plotting *Q_e_* versus *C_e_*.

To evaluate the adsorption kinetics, a pseudo-first-order kinetic equation and a pseudo-second-order kinetic equation were employed to analyze the adsorption kinetics of Cr(VI) [29]. The pseudo-first-order equation (Equation (2)) and pseudo-second-order kinetic equation (Equation (3)) are given as follows:(2)$$\ln(Q_{e}-Q_{t})=\ln Q_{e}-k_{1}t$$
(3)$$\frac{t}{Q_{t}}=\frac{1}{k_{2}Q_{e}^{2}}+\frac{t}{Q_{e}}$$
(4)$$h=k_{2}Q_{e}^{2}$$
where *t* (min) is the adsorption time, *Q_t_* (mg/g) is the adsorption amount at given time *t*, and *Q_e_* (mg/g) is the equilibrium adsorption capacity; *k_1_* (min^−1^) and *k_2_* (g/(mg min)) are the rate constants of the pseudo-first-order and pseudo-second-order adsorption kinetic models. In particular, *h* (mg/(g min)) is the initial adsorption rate.

Freundlich (Equation (5)) and Langmuir (Equation (6)) isotherm equations were fitted to experimental results in this work. The Freundlich model is a multilayer adsorption isotherm for surface inhomogeneities. The nonlinear form of the Freundlich model is as follows:(5)$$Q_{e}=k_{f}C_{e}^{1/n}$$
where *k_f_* is the Freundlich constant which indicates the relative adsorption capacity of the adsorbent; *1/n* is the Freundlich constant indicating the intensity of adsorption.

The Langmuir model is a monolayer adsorption isotherm model for the homogeneity surface. The nonlinear form of the Langmuir model is expressed as follows:(6)$$Q_{e}=\frac{Q_{m}K_{L}C_{e}}{1+K_{L}C_{e}}$$
where *K_L_* (L/mg) is the Langmuir constant which is related to the energy of adsorption and the affinity of the binding site interacting with adsorbates. *Q_m_* (mg/g) is the Langmuir constant indicating the maximum adsorption capacity of the adsorbent.

## 3. Results

### 3.1. Characterization of Adsorbent

The morphology and structure of APNs and nanofibers were studied by SEM and TEM, and the corresponding images are shown in Figure 1. In the images of APNs (Figure 1a,b), monodisperse nanospheres with a diameter of about 500 nm were clearly observed. The pure PAN nanofibers presented uniform size and smooth surfaces after the electrospinning process (Figure 1c). According to the enlarged image (Figure 1d), the average diameter of PAN nanofibers was 640 nm. With the addition of APNs, nanofibers became thicker with a diameter increased to 2600 nm (Figure 1e,f). A rougher surface was formed with APNs, indicating the successful loading of APNs in PAN nanofibers.

Nitrogen adsorption–desorption isotherms were used to evaluate the porosity of the structure. As shown in Appendix A, nearly no nitrogen was adsorbed on pure PAN nanofibers. Among APNs and nanofibers, APNs presented the highest porosity with the largest surface area and pore volume. All the samples exhibited the typical type I isotherm, indicating a mainly microporous structure. After the electrospinning process, surface area and pore volume both decreased by about one-third, which was consistent with the mass ratio of APNs in PAN/APNs.

The chemical properties of APNs and nanofibers were further examined by FTIR spectra. As shown in Figure 2a, a broad band centered at 3400 cm^−1^ was observed on APNs and could be identified as the stretching vibrations of O-H and N-H [26]. The band around 1640 cm^−1^ was assigned to -NH- in APNs [27]. Although the intensity was weakened, the above typical bands were still present in the FTIR spectra of PAN/APN nanofibers after the electrospinning of APNs. These results furthermore indicated that APNs were incorporated into PAN nanofibers. The zeta potential of APNs was tested at different pH levels (from 1 to 13) to analyze the changes in electricity. As shown in Figure 2b, the pH_PZC_ of APNs was nearly 9.1, which means APNs were positively charged in acidic and weakly basic media.

### 3.2. Adsorption Kinetic

The effect of contact time on the removal behavior of Cr(VI) for APNs, PAN/APNs and pure PAN nanofibers is shown in Figure 3. It is shown that APNs could remove 88% of Cr(VI) within 1 min and reach equilibrium in the next 10 min. The ready and fast adsorption of toxic Cr(VI) was ascribed to short intraparticle diffusion distance and the high density of active binding sites on APNs [30]. For pure PAN nanofibers, the adsorption equilibrium could be reached within 15 min. However, poor adsorption removal for Cr(VI) was detected due to its feeble surface interaction with Cr(VI). With the addition of APNs, PAN/APN nanofibers removed 40% of Cr(VI) in 1 min and further achieved adsorption equilibrium within 120 min. Although the adsorption time of PAN/APNs was longer than that of APNs, the adsorption capacity of Cr(VI) was comparable in equilibration time. As shown in Appendix A, the adsorption capacities of APNs, PAN/APNs and PAN nanofibers were 98.3, 97.3 and 15.7 mg/g, respectively. The result indicated that the loading of APNs in PAN nanofibers endowed PAN/APNs with high adsorption capacity.

In addition, pseudo-second-order kinetic (Appendix A) and pseudo-first-order kinetic (Appendix A) models were employed to study adsorption kinetics. The relative kinetic parameters are also summarized in Appendix A. According to the simulated equilibrium adsorption capacities, the results of pseudo-second order were more reliable with highly consistent with the experimental capacities when compared with pseudo-first order. The values of correlation coefficients clearly showed more precise fitting using the pseudo-second-order kinetic model, implying that the adsorption rate for Cr(VI) was controlled by chemical reaction [31].

### 3.3. Adsorption Isotherms and Thermodynamics

The adsorption isotherms of APNs, PAN/APNs and PAN nanofibers are shown in Figure 4. The maximum adsorption capacities (Q_m_) of APNs at 298, 308 and 318 K were 698, 781 and 922 mg/g, respectively. For PAN/APN nanofibers, Q_m_ values were 556, 667, and 800 mg/g at 298, 308 and 318 K, respectively. Furthermore, the adsorption capacity of PAN nanofibers is listed in Appendix A. It can be calculated that APNs in PAN/APN nanofibers maintained about 96.5% of their adsorption capacity after the contribution of the PAN substrate was excluded. In order to evaluate the adsorption performance for Cr(VI), the relevant results for organic nanomaterials and their derivatives are compiled in Table 1 [19,21,31,32,33,34,35,36]. It can be noticed that APNs and PAN/APN nanofibers presented comparable adsorption capacities to others, which revealed that APNs incorporated in PAN nanofibers could be promising adsorbents for the removal of Cr(VI).

The Langmuir isotherm and Freundlich isotherm were employed to describe the adsorption data (Figure 4). The corresponding parameters and their correlation coefficient R^2^ values are also shown in Appendix A. Compared with the Freundlich model, better fitness with a higher R^2^ value was founded for the modeling of adsorption isotherms using the Langmuir model. The saturation adsorption capacities of adsorbents calculated from the Langmuir isotherm model approached the experimental data. This also indicated that the adsorption of Cr(VI) on adsorbents follows a monolayer adsorption process rather than a multilayer adsorption process. Meanwhile, it also reflected that the specific active sites were homogeneously distributed onto the surfaces of APNs and PAN/APN nanofibers.

To understand the adsorption performance from the perspective of thermodynamics, the relative parameters enthalpy change (∆*H*), free energy change (∆*G*) and entropy change (∆*S*) were calculated using the following equations:(7)$$\Delta G=-RT\ln K_{c}$$
(8)$$\ln K_{c}=-\frac{\Delta H}{RT}+\frac{\Delta S}{R}$$
(9)$$K_{c}=\frac{Q_{e}}{C_{e}}$$
where *R* (8.314 J/(mol K)) is the gas constant; *T* (K) is the absolute temperature; and *K_C_* (L/g) is the adsorption equilibrium constant, which was obtained from the Langmuir isotherm at different temperatures.

The values of ∆*G*, ∆*H*, *K_c_* and ∆*S* were calculated and are presented in Table 2. A positive value of ∆*H* suggested that the reaction between the adsorbent and adsorbate involved an endothermic process. A positive value of ∆*S* indicated that Cr(VI) was adsorbed onto the internal part of the adsorbent, increasing the randomness at the solid–solution interface [36]. A negative value of ∆*G* at different temperatures indicated that the removal of Cr(VI) by APNs and PAN/APNs follows a spontaneity process. With the increase in temperature, ∆*G* became more negative, manifesting that higher temperature could facilitate the adsorption of Cr(VI) on APNs and PAN/APN nanofibers. Hence, a higher temperature was a favorable factor for the adsorption procedure.

### 3.4. Effect of pH and Ionic Strength

Initial pH is an important parameter that affects the adsorption efficiency for Cr(VI). For the adsorbent and adsorbate, pH has a strong impact on the surface charge of functional groups and the ionic and charge states of active sites on the adsorbent, as well as Cr(VI) species [29]. Figure 5a shows the effect of the initial pH in an aqueous environment on Cr(VI) adsorption onto PAN/APN nanofibers. The adsorption capacity of Cr(VI) on PAN/APN nanofibers increased dramatically as pH decreased. Cr(VI) exists in five main species, namely H_2_CrO_4_, HCrO_4_^−^, CrO_4_^2−^, HCr_2_O_7_^−^ and Cr_2_O_7_^2−^, in aqueous solutions [37]. When pH > 6, CrO_4_^2−^ is the primary species. When pH is 2–6, HCrO_4_^−^ and Cr_2_O_7_^2−^ are the predominant species. H_2_CrO_4_ is the main species existing in an aqueous solution when pH < 1 [11]. At low pH, a positively charged surface and high protonation were realized on PAN/APN nanofibers, which favored the electrostatic interaction of Cr(VI) anions, leading to remarkable adsorption performance [38]. Along with the physical adsorption, the chemical reduction process is also involved in the removal of Cr(VI) on PAN/APN nanofibers. The oxidizability of Cr(VI) in acidic and alkaline conditions is represented in the following reaction equations, respectively [19]:(10)$$HCrO_{4}{}^{-}+7H^{+}+3e\to Cr^{3+}+4H_{2}O,\;E_{0}=+1.35\;V$$
(11)$$CrO_{4}{}^{2-}+2H_{2}O+3e\to CrO_{2}{}^{-}+4OH^{-},\;E_{0}=-0.12\;V$$

As shown in the equations, when H^+^ is involved in the reaction, Cr(VI) can easily be reduced to Cr(III), indicating that lower pH could benefit the conversion of Cr(VI) to Cr(III). In alkaline conditions, it was difficult for the reduction of main anion species CrO_4_^2-^ to occur due to the quite low reduction potential. The result can be verified by the concentration of Cr(III) in the equilibrium solution after adsorption. Figure 5b shows that Cr(III) was barely detected under high pH. With an increase in pH, the decreased adsorption capacities could be attributed to the surface charge state shift from positive to negative, increasing the repulsive force between Cr(VI) and PAN/APNs. Additionally, in strongly alkaline solutions, the abundant OH^-^ competes with Cr(VI) anions, which also results in poor adsorption capacity [30]. These results suggested that low pH is beneficial to PAN/APN nanofibers removing Cr(VI).

The coexisting ions were another parameter that would affect the adsorption efficiency of Cr(VI) in the solution. The competitive influence of foreign ions such as Na^+^, Ca^2+^, K^+^, SO_4_^2−^, Cl^−^, NO_3_^−^ and PO_4_^3−^ was investigated. As shown in Figure 6, ions Na^+^, Ca^2+^, K^+^, Cl^−^, NO_3_^−^ and PO_4_^3−^ had little competitive influence on the adsorption process of Cr(VI) on PAN/APN nanofibers. When SO_4_^2−^ was introduced, the removal efficiency of Cr(VI) clearly declined, indicating a passive impact on Cr(VI) adsorption [29,38]. The existing SO_4_^2−^ remarkably decreased adsorption capacity of Cr(VI) owning to several factors. The different effect of coexisting anions was dependent on the affinities between the adsorbent and the coexisting anions. The interaction between the amino functional group and SO_4_^2−^ was higher than that for other anions. Therefore, the force of affinity between PAN/APNs and SO_4_^2−^ was stronger than that for other coexisting ions. SO_4_^2−^ in solution would occupy surface sites to decrease the availability of active sites on the adsorbent, which was the main reason for the degradation of adsorption performance. In addition, the decreased surface charge of the adsorbent increased the electrostatic repulsion between the Cr(VI) anion and the adsorbent. Moreover, SO_4_^2−^ and Cr(VI) could form a kind of strong complex, which caused lower affinity for the adsorbent surface active sites [39].

### 3.5. Cr(VI) Removal Performance in Bed Column

The Cr(VI) removal performance of PAN/APN nanofibers in a continuous fixed-bed adsorption column was also investigated to examine their practical application in wastewater treatment. The corresponding results are shown in Figure 7. The breakthrough point of Cr(VI) was set as 0.2 mg/L, which was restricted by “GB 39731-2020 Discharge standard of water pollutants for electronic industry” of China. In Figure 7a, the breakthrough curve shows that PAN/APN nanofibers can deal with nearly 4600 BV Cr(VI)-contaminated water. It is worth noticing that the concentration of Cr(VI) in 4325 BV effluent was below 5 μg/L. After the desorption cycle, PAN/APN nanofibers maintained good performance for Cr(VI) removal. The column adsorption results indicated that PAN/APN nanofibers could be a potential adsorbent for water treatment and reuse.

### 3.6. The Possible Mechanism of Cr(VI) Removal

To clarify the mechanism of Cr(VI) removal, various techniques were applied to investigate the adsorption process. Figure 8 shows SEM and EDS images of PAN/APNs before and after Cr(VI) adsorption. The mapping of C (Figure 8c,i), N (Figure 8d,j) and O (Figure 8e,k) signals showed that these signals were present for both PAN/APNs and PAN/APNs-Cr. The mapping of the Cr (Figure 8l) signal only showed the presence of this signal in PAN/APNs-Cr. The signal of Cr was evenly distributed on the surface of the nanofibers and adhered with nitrogen species (Figure 8h,l), which indicated that N components were the active sites that interacted with chromium. To verify this conjecture, XPS and FTIR were further applied to analyze the structure variation of the adsorbent (Figure 9).

The wide XPS spectra of APNs-Cr, PAN/APNs-Cr and PAN-Cr are given in Figure 9a. The typical Cr peak appeared, indicating the existence of Cr on the surface of adsorbents. The Cr 2p spectra of APNs-Cr and PAN/APNs-Cr (Figure 9b) could be deconvoluted to two peaks at about 577.0 eV and 578.7 eV for Cr 2p_3/2_ and Cr 2p_1/2_ which correspond to the oxidation state of Cr(III) and reduction state of Cr(VI), respectively [19,31,34]. The Cr(III) peak verified that the removal of Cr(VI) involved a reduction process. However, in the Cr 2p spectra of PAN-Cr, Cr(VI) could only be detected at 583.3 eV. The results indicated that the reduction of Cr(VI) to Cr(III) and adsorption of Cr(III) concurrently happened on APNs.

To simplify this issue, the N 1s spectra of APNs and APNs-Cr were analyzed carefully (Figure 9c). Two kinds of nitrogen species, i.e., pyrrolic-type nitrogen (-NH-, N_py_) at 400.2 eV and pyridinic-type nitrogen (-N=, N_pr_) at 399.2 eV, were deconvoluted on APNs, and they can coordinate with Cr(III)-form chelate [27]. The molar percentages of N_py_ and N_pr_ were 42.7 and 57.3%, respectively. After the adsorption process, there was a new amine species of protonated quinoid imine (N^+^_py_) detected at 401.3 eV. The deconvolution of N 1s spectra on APNs-Cr resulted in the assignment of three peaks: 53.8% of N_pr_, 29.8% of N_py_ and 16.4% of N^+^_py_. The sharp decrease in the molar ratio of N_py_ implied that N_py_ was oxidized to N^+^_py_ by Cr(VI) [21,34]. Oxidized states and neutral reduction states shuttle back and forth in a reversible process. Therefore, during Cr(VI) removal, Cr(VI) may be reduced to Cr(III) via spontaneous electron transfer from N_py_ to Cr(VI). In an acidic medium, the reduction of Cr(VI) can be described by the following reaction [38]:(12)$$Cr_{2}O_{7}{{}^{2-}}_{\left(aq\right)}+6N_{py(s)}+14H^{+}{}_{\left(aq\right)}\to 2Cr^{3+}{}_{\left(aq\right)}+6N^{+}{}_{py(s)}+7H_{2}O_{\left(l\right)}$$
(13)$$N^{+}{}_{py(s)}+e\to N_{py(s)}$$

The formal potential of Cr_2_O_7_^2-^/Cr^3+^ was 1.35 V, and the corresponding value for N^+^_py(s)_/N_py(s)_ was around −0.2 V. The standard free energy of Equation (12) was −448.66 kJ/mol. These results indicated that the redox reaction is spontaneous.

In addition, the molar ratio of N_pr_ decreased by 3.5% after the Cr(VI) adsorption process. This indicates that N^+^_py_ might be partly produced via doping of H^+^ on N_pr_. Moreover, the existence of N^+^_py_ favored the electrostatic interaction with negative Cr(VI) species to promote the adsorption behavior [22].

In FTIR spectra of APNs-Cr (Figure 9d), the peak around 1640 cm^−1^ corresponded to -NH-. Compared with the spectra of APNs (Figure 2), the relative intensity declined after adsorption [32], further proving the redox reaction between APNs and Cr(VI). Meanwhile, two new peaks appeared near 950 and 790 cm^−1^, corresponding to characteristic peaks of Cr=O and Cr-O, respectively [20]. These signals could also verify the electrostatic interaction of the Cr(VI) anion with the adsorbent in the adsorption process.

Based on the above analysis and discussion, the mechanism of Cr(VI) removal by APNs and PAN/APN nanofibers could be speculated as occurring in two stages: (i) Cr(VI) anions rapidly diffused to the surface of the adsorbent and were bound onto APNs and PAN/APN nanofibers due to electrostatic interaction between protonated N^+^_py_ and negative charged Cr(VI) species; (ii) Cr(VI) anions relatively slowly diffused to the interior resin and were simultaneously reduced to less toxic Cr(III) and further chelated with N_pr_ imino groups.

## 4. Conclusions

In summary, PAN/APN nanofibers were successfully prepared by electrospinning of monodispersed APNs. The structure and property results illustrated that the obtained nanofibers present high-density nitrogen-containing functional groups. Based on the adsorption of toxic Cr(VI), the adsorption capacity reached 556 mg/g at 298 K, exhibiting excellent Cr(VI) removal ability. The dynamic adsorption results showed that PAN/APN nanofibers can deal with 4600 BV Cr(VI)-contaminated water under the discharge standard of 0.2 mg/L. The coexisting ions such as Na^+^, Ca^2+^, K^+^, Cl^−^, NO_3_ and PO_4_^3−^ had little influence on Cr(VI) removal efficiency, while SO_4_^2−^ individually restricted the Cr(VI) adsorption. The adsorption process not only involved the electrostatic attraction between the surface of PAN/APN nanofibers and Cr(VI) anions but also involved a redox reaction, in which Cr(VI) was reduced to low-toxicity Cr(III) by N_py_, followed by Cr(III) chelation on the surface of absorbent N_pr_ imino groups. The adsorption process was spontaneous and endothermic. All of these results indicated that PAN/APN nanofibers have a great potential to remove Cr(VI) from wastewater.

## Figures and Tables

**Figure 1 molecules-27-07133-f001:**
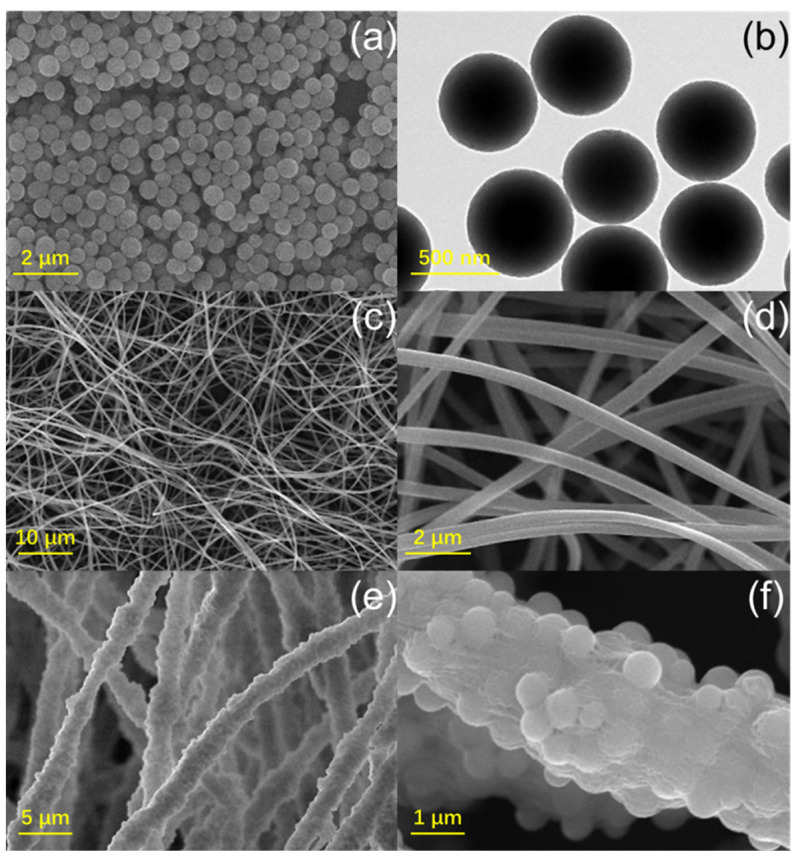
SEM (**a**) and TEM (**b**) images of APNs; SEM images of PAN (**c**,**d**) and PAN/APNs (**e**,**f**).

**Figure 2 molecules-27-07133-f002:**
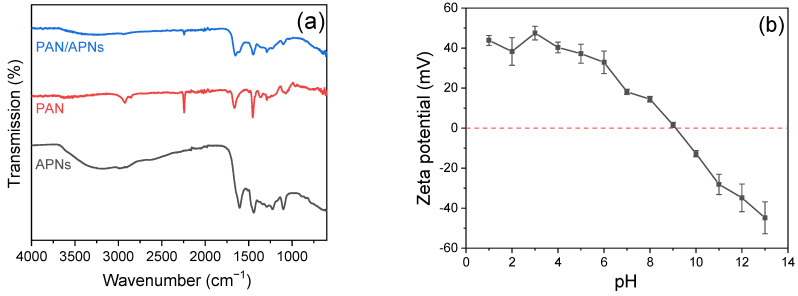
FTIR spectra of APNs, PAN and PAN/APNs (**a**) and zeta potential of APNs (**b**).

**Figure 3 molecules-27-07133-f003:**
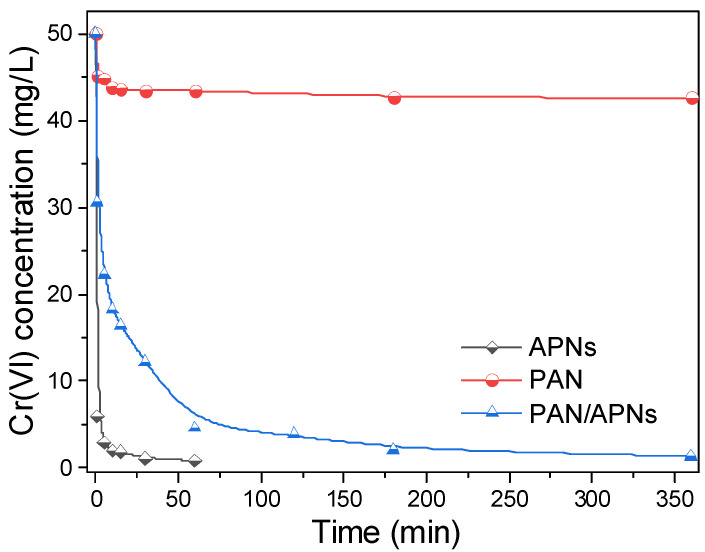
Adsorption rate of Cr(VI) by APNs, PAN/APNs and PAN (C_0_ = 50 mg/L; adsorbent dose = 0.5 g/L; T = 298 K; pH = 2).

**Figure 4 molecules-27-07133-f004:**
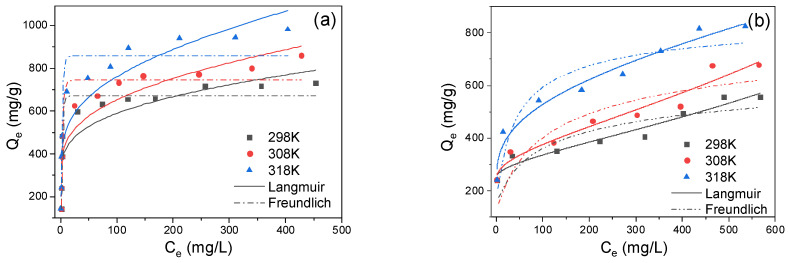
Langmuir and Freundlich isotherm models for Cr(VI) adsorption onto APNs (**a**) and PAN/APNs (**b**) (pH = 2, adsorbent dose = 0.2 g/L).

**Figure 5 molecules-27-07133-f005:**
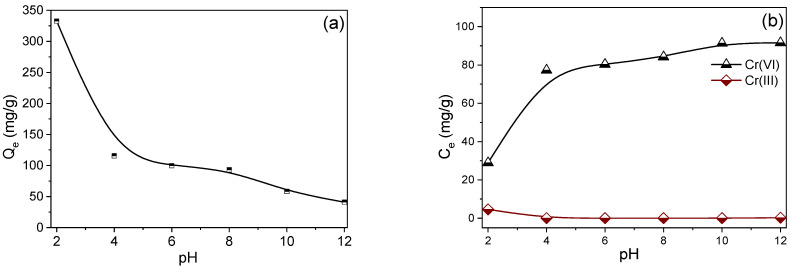
Effect of pH on Cr(VI) adsorption onto PAN/APNs (**a**); concentration of Cr(VI) and Cr(III) after adsorption on PAN/APNs (**b**) (T = 298 K, adsorbent dose = 0.2 g/L, C_0_ = 100 mg/L).

**Figure 6 molecules-27-07133-f006:**
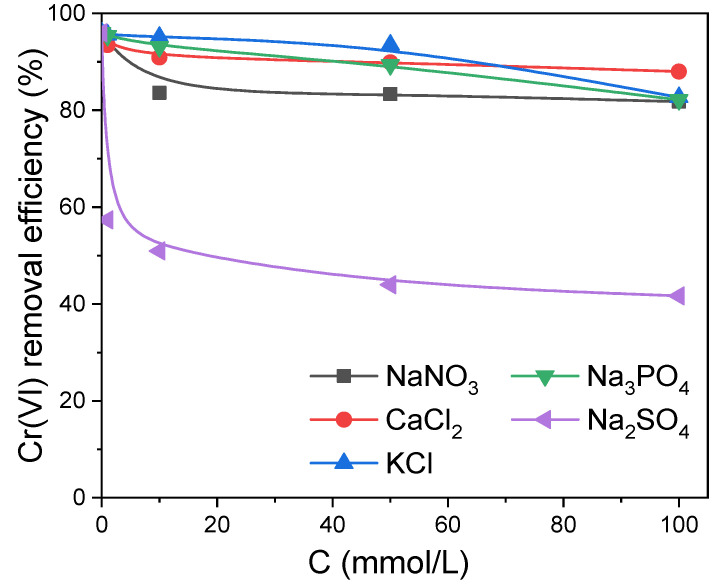
Effect of foreign ions and their ionic strength for Cr(VI) adsorption onto PAN/APNs (T = 298 K, pH = 2, adsorbent = 0.2 g/L, C_0_ = 50 mg/L).

**Figure 7 molecules-27-07133-f007:**
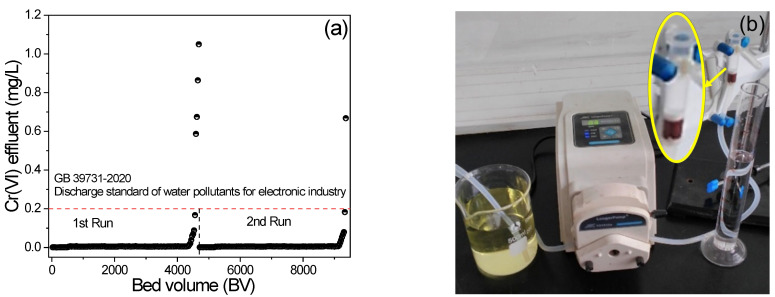
Breakthrough curves for adsorption of Cr(VI) onto PAN/APNs (**a**); fixed-bed column experiments (**b**) (fixed bed was the tube in enlarged image; bed volume = 0.78 cm^3^, C_0_ = 10 mg/L, flow rate = 30 BV/h, pH = 2).

**Figure 8 molecules-27-07133-f008:**
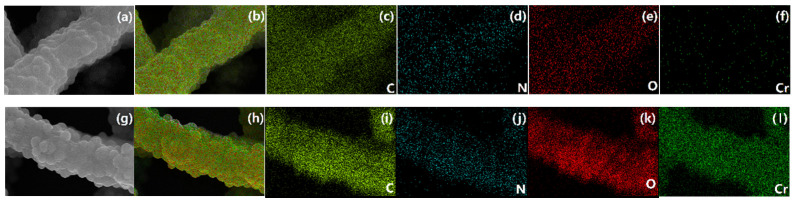
SEM and EDS images of PAN/APNs (**a**–**f**) and PAN/APNs-Cr (**g**–**l**): SEM (**a,g**), overlapped EDS image (**b,h**), carbon (**c,i**), nitrogen (**d,j**), oxygen (**e,k**) and chromium (**f,l**).

**Figure 9 molecules-27-07133-f009:**
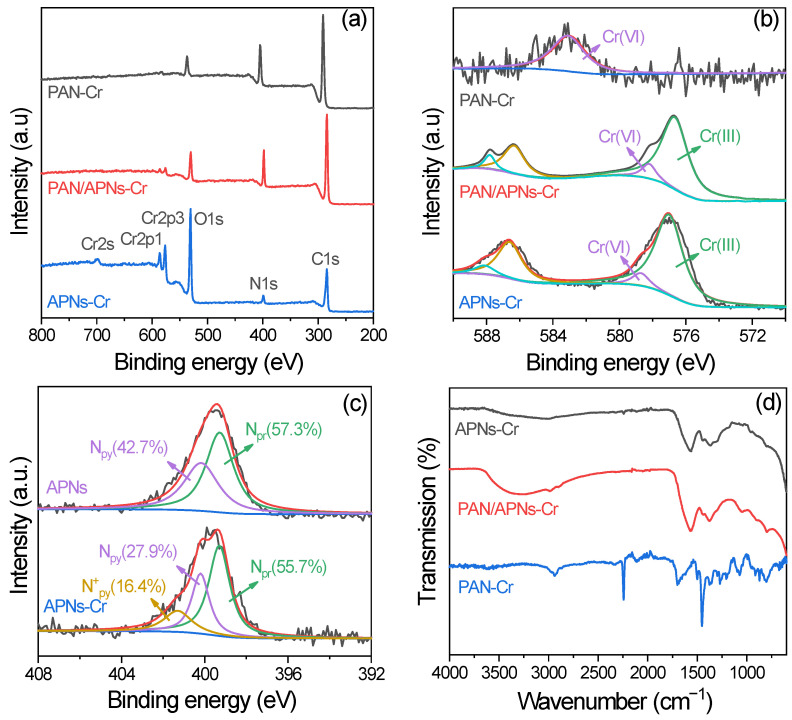
XPS survey scan spectra (**a**), high-resolution spectra of Cr 2p (**b**) and N1s (**c**) and FTIR spectra (**d**) of adsorbents after adsorption.

**Table 1 molecules-27-07133-t001:** The Q_m_ values of APNs, PAN/APNs and other reported adsorbents.

Adsorbents	Q_m_ (mg/g)	Reference
Polypyrrole/GO	625	[33]
G-PDAP/GO	609.76	[19]
Poly(m-phenylenediamine)	500.00	[32]
Magnetic PS-EDTA resin	250.00	[34]
Fe_3_O_4_@poly(m-Phenylenediamine)	246.09	[21]
Diethylene triamine grafted glycidyl methacrylate	143	[35]
Hydrophobic resin with tricaprylmethylammonium chloride	71.24	[36]
PEI-facilitated ethyl cellulose	36.8	[31]
APNs	698	This work
PAN/APNs	556	This work

**Table 2 molecules-27-07133-t002:** Thermodynamic parameters for Cr(VI) adsorption on APNs and PAN/APNs.

Samples	*T* (K)	*Kc* (L/g)	∆*G* (kJ/mol)	∆*H* (kJ/mol)	∆*S* (J/(mol K))
APNs	298	1.61	−1.16	16.32	59.00
	308	2.00	−1.77		
	318	2.43	−2.34		
PAN/APNs	298	1.14	−0.315	19.67	67.05
	308	1.45	−0.959		
	318	1.87	−1.656		

## Data Availability

The authors confirm that all data underlying the finding are fully available without restriction. Data can be obtained after submitting a request to the corresponding/first author.

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
