# Peer review of "Polyacrylonitrile/Aminated Polymeric Nanosphere Nanofibers as Efficient Adsorbents for Cr(VI) Removal"

_molecules, 2022, doi:10.3390/molecules27207133_

Round 1

Reviewer 1 Report

This manuscript reported the production route ofpolyacrylonitrile/aminated polymeric nanospheres (PAN/APNs) nanofibers and its application on Cr(VI) adsorption. A substantial amount of work has been done, and some findings were interesting. I am recommending this work to be considered for publication in “Molecules” after undergoing minor revision. The reviewer genuinely hopes that the comments help improve the value of the manuscript.

1. Page 2 Line 86-92: How does the authors determine the ingredient ratio in the preparation process of PAN/APNs. It should be noted that the APNs loading has an important influence on the adsorption properties of PAN/APNs.

2. Page 4 Line 140: The authors are required to provide the pore structure data of the adsorbents.

3. Page 7 Line 221: Thermodynamic parameters for Cr(VI) adsorption on APNs are preferred.

4. Page 6 Line 183: According to the kinetics results, the authors stated that the adsorption rate for Cr(VI) is controlled by chemical reaction. However, the authors also stated that Cr(VI) removal was primarily dominated by physical adsorption based on thermodynamics and competitive adsorption study. The adsorption mechanism should be carefully described.

5. Page 9 Line 268-270: Sulfate was considered to have an adverse effect on Cr(VI) adsorption. How to improve the adsorption selectivity of PAN/APNs?

6. Page 11 Line 320: It’s not appropriate to denote quaternary nitrogen or protonated quinoid imine by –N+=.

7. To further verify the electrostatic interaction mechanism in Cr(VI) adsorption process, the Zeta potential variations should be provided.

Reviewer 2 Report

This paper reported a preparation method of polyacrylonitrile/aminated polymeric nanospheres to adsorb Cr(VI) from aqueous solution. The material was characterized in detail. The adsorption kinetics and the effects of other ions were both obtained. The mechanisms of reduction and chelation were interpreted. So, in my opinion, this is a good manuscript which should be accepted after answering the following questions.

1. Tenses problem. The past formula is more suitable for describing the completed test, the working conditions used in the test and the test results.

2. Figure 7. No bed was found in Fig. 7b which should be introduced in section 2.Materials and Methods.

3. Line 170. "Listed in Table S1, xxxx" should be "As shown in Table S1". 

Reviewer 3 Report

1. The writing can be improved, and grammatical errors need to be polished.

2.please mention the type of statistical analysis performed on the measured data. 

3. What is the main separation mechanism of Cr from the solution.

4. How is the composite designed in this work? Justification could be useful.

5.I suggest improve introduction section as follow. Please bold the importance of heavy metal quantification further their removal briefly, you can read and cite:

a)https://doi.org/10.2166/wrd.2016.073

b) https://doi.org/10.1016/S1003-6326(16)64341-8

6. In experimental section, please write clearly all amounts (in mmole), concentrations, volumes used for synthesis procedures and avoid mentioning the gram alone. Please check all. This section should be easily applicable by readers. All procedures used from literature should be cited.

7. Authors should be determine and analyze pHpzc of the adsorbent.

8. In isotherm section please enhance discussion.

9. Can you any evidence for the type of the removal process, ion exchange, chemi or physiosorption?

Round 2

Reviewer 3 Report

All items are OK.